# SAGE: Adaptive Self-Training Towards Endogenous Memory in Large Language Models

## Abstract

Large language models exhibit strong generalization but face limitations in real-world adaptation, as their parameters remain static. Inspired by neuroscience and cognitive science, this work investigates endogenous memory as a mechanism for adaptive and incremental updates. We present SAGE, a framework for self-adaptive parameter updates in reasoning tasks, triggered by the detection of out-of-distribution knowledge. SAGE consists of three core modules: (1) a Trigger module, which detects reasoning failures across multiple evaluation metrics in real time; (2) the Trigger Buffer module, which clusters reasoning failure samples using a streaming clustering process, followed by stability checks and similarity-based merging; and (3) the LoRA Store module, which dynamically optimizes parameter updates with an adapter pool for knowledge retention. Evaluation results show that SAGE demonstrates excellent accuracy, robustness, and stability on the atomic reasoning subtask through dynamic knowledge updating during test time. Specifically, an EM accuracy of $97.16\%_{\pm 4.65\%}$ reflects statistically significant and reliable performance.

## 1 Introduction

Large language models (LLMs), pre-trained on massive corpora, exhibit remarkable generalization. However, LLMs no longer adapt in real-world interactions, as their parameters remain fixed. Toward true artificial general intelligence, LLMs are expected to extract useful information from the environment and integrate it to enhance their capabilities. Therefore, investigating endogenous memory mechanisms in LLMs is crucial. Inspired by neuroscience and cognitive science, long-term memory that influences cognition can be categorized into two types: emotion-related memory and memory of learned out-of-distribution (OOD) knowledge. Given the similarity between knowledge memory and LLM pre-training tasks, as well as the need to expand OOD knowledge, this paper addresses the core question: *How can LLMs adaptively and incrementally update under OOD condition?*

Endogenous memory in LLMs allows models to modify cognition, which inherently requires parameter updates. However, research has historically focused mainly on external memory. Early studies showed that the transformer feedforward network (FFN) exhibits key-value (KV) storage (Geva et al., 2021), prompting the integration of KV pairs into FFNs (Wu et al., 2022; Berges et al., 2024; Li et al., 2025). Explicit memory (Yang et al., 2024) inspired retrieval-augmented generation (Lewis et al., 2020; Liu et al., 2024) to use external cues for memory formation (Gutiérrez et al.; Jimenez Gutierrez et al., 2024). Other approaches include long-context augmented input (Wang et al., 2023) and retrieval augmentation (Borgeaud et al., 2022) to improve persistence of internal knowledge. However, these methods only temporarily expand knowledge without enhancing integration. MemoryLLM (Wang et al., 2024) and related self-update models (Huang et al.; Shafayat et al., 2025) have advanced memory mechanisms, but most focus on self-evolution after real-world datasets are exhausted, rarely addressing updates at knowledge boundaries.

To this end, we propose **SAGE**, a large model endogenous memory framework triggered by detecting OOD knowledge, which aims to enable self-adaptive parameter updates of LLMs on reasoning tasks. SAGE implements three core functions: (1) detection of reasoning failures, (2) clustering of abnormal samples, and (3) dynamic optimization of parameter updates. The Trigger module evaluates model outputs across multiple dimensions (model confidence, model behavior, and semantic similarity) to detect reasoning failures in real time, using threshold gating to further distinguish

them. The Trigger Buffer module then clusters these anomalous samples. It firstly buckets them to reduce domain interference and matches each sample to a similar cluster. It then performs clustering based on the number of arriving samples and merges clusters according to embedding similarity. Finally, the Lora Store module maintains a dynamic pool of parameter-efficient adapters. It initially searches through the parameter space (e.g., rank, learning rate, dropout) for adapter training and ranks the top-$k$ candidate configurations by accuracy and cross-entropy loss. It then performs local expansion optimization to retain the top-3 adapters for reuse in the future.

In the experiments, each module was evaluated as follows: the Trigger module was assessed through false positive rate detection, threshold sensitivity analysis, and indicator weight sensitivity analysis. The results demonstrate the module's strong discriminative power in OOD datasets, validating the effectiveness of the selected indicators. Specifically, approximately 57% of the plateaus occurred within the threshold range, while 62.5% occurred within the indicator weight range, indicating robust accuracy under varying conditions. For the Trigger Buffer module, ablation studies on HDB-SCAN, stability checks, and merging confirmed the design's validity. Dynamic visualization and sensitivity analysis of stream data clustering demonstrate reliable performance with streaming data. The evaluation of the LoRA Store module focused on fine-tuning accuracy. In atomic task evaluations, it outperformed reasoning with multitask datasets. Additionally, heatmap tests of LoRA parameter rank and learning rate show significant impact on fine-tuning accuracy (up to 83%). After combining the three modules, the EM accuracy of the SAGE framework increased from 81.91% to 94.85%, while the MAE and MSE decreased by several orders of magnitude, demonstrating enhanced robustness and stability. Finally, the ablation study further validated SAGE's design and its effectiveness in supporting reasoning during inference.

## 2 PRELIMINARIES

**Problem Statement**   The core challenge addressed in this work is *designing self-adaptive LLMs capable of continuously incorporating new inputs, ensuring robust knowledge integration despite limited supervision and sparse data*. Therefore, we formulate the requirements for self-adaptive LLMs: **Adaptation scope**: The model should *perform localized parameter updates*, avoiding large-scale parameter updates. **Adaptation phase**: Training must occur *during test-time interaction*, as the model engages with users in real-time to incorporate new knowledge, rather than during post-training or pre-training of LLMs. **Adaptation objective**: The model must ensure that *updated parameters preserve accuracy and robustness*, preventing undesirable drift in global parameters. **Adaptation autonomy**: The model should *autonomously determine when and how to adapt*, selecting relevant instances from interactive data streams, while independently optimizing algorithms and hyperparameters. **Adaptation challenges**: The model mustfunction in environments where data is *noisy, limited, and unordered*, with incoming tasks that are open-ended and dynamically evolving.

**Sketch of SAGE**   To address the challenges of self-adaptive LLMs, we propose a trigger-guided self-adaptation framework. Our key insight is to reduce adaptation complexity by *decomposing complex reasoning tasks into atomic subtasks*, which are minimal and independent. This decomposition enables the model to adapt more easily, reduces error accumulation, and ensures stable updates for improved accuracy. Based on this insight, we introduce **SAGE**: **Trigger**, which converts static fine-tuning into dynamic adaptation by detecting reasoning failures on OOD inputs and determines whether the data should be retained for future adaptation; **Trigger Buffer**, which caches and cluster anomalous samples to improving the quality of subsequent fine-tuning by increasing inter-sample coherence; and **LoRA Store**, which performs parameter-efficient fine-tuning on stable clusters, with parameters automatically adjusted based on performance. The most effective adapters are preserved for future reuse, ensuring efficient adaptation in similar tasks.

## 3 DESIGN OF SAGE

### 3.1 TRIGGER

The trigger module is responsible for initiating dynamic adaptation by detecting LLM reasoning failures. Several studies have explored how LLMs know when they don't know, either through consistency across multiple responses (Shafayat et al., 2025; Zuo et al., 2025; Prasad et al.) or

through hidden state probe training (Chen & Varoquaux, 2025). However, these approaches have achieved limited accuracy. Given the objectivity of knowledge memorization tasks, we considered using ground truth answers to enable the LLM to self-check whether it has mastered a certain type of knowledge. As shown in Figure 1, our Trigger module consists of two components: response scoring and threshold gating.

**Response Scoring.**  To improve detection accuracy, we use three evaluation levels: token-level confidence from logits (*Logits Margin*), model behavior at the token level (*BLEU* and *ROUGE-L*), and semantic similarity from embeddings (*Embedding Similarity*). As shown in Figure 1, after the input $x^t \in \mathbb{R}^d$ reaches time $t$ passes through the LLM, implemented as a transformer decoder, the margin score, Logits Margin (LM) (Liu et al., 2016), is calculated based on the logits output from the Linear layer in Figure 1 to evaluate the confidence in the prediction of the next token. Assume that the logits of the model output form a vector $z \in \mathbb{R}^V$, where $V$ is the vocabulary size. Then we have,

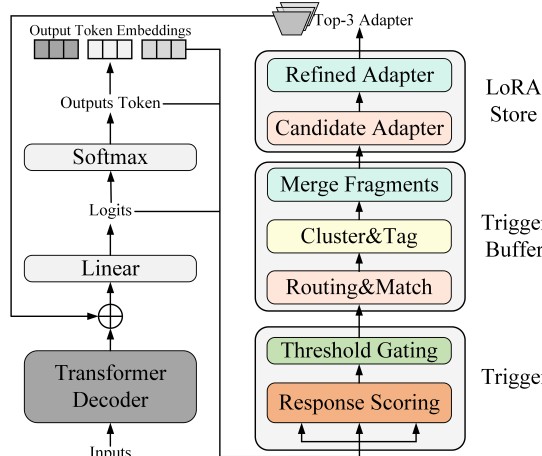

Figure 1: The architecture of SAGE.

$$LM = z_{(1)}(x^t) - z_{(2)}(x^t) \qquad (1)$$

The next token obtained by applying the softmax function in Figure 1 is treated as the predicted token, which is then compared against the ground truth. (Note: During training, the LLM receives a masked version of the golden answer and retains it as the ground truth answer.) We use BLEU (Papineni et al., 2002) for local exact matching and ROUGE-L (Lin, 2004) for global sequence matching. Specifically, we adopt a variant of BLEU-4 to measure the 4-gram overlap between the predicted and ground truth answers, defined using 4-gram precision $p_4$ and weights $w_4$:

$$B_4 = \exp\left(\sum_{n=1}^{N} w_4 \log p_4\right) \qquad (2)$$

ROUGE-L, on the other hand, measures the longest common subsequence regardless of contiguity. We use the $F_1$ score, computed as the harmonic mean of precision $P$ and recall $R$, and define it,

$$F_1 = \frac{2 \cdot P \cdot R}{P + R} \qquad (3)$$

Finally, the predicted answer and the ground truth answer are encoded into embeddings in Figure 1 to measure their semantic similarity. Let $\mathbf{Emb}_{pred}(x^t)$, $\mathbf{Emb}_{true}(x^t)$ denote the embedding vectors. The embedding similarity (Lin et al., 2017) $ES$ is,

$$\text{ES} = \frac{\mathbf{Emb}_{pred}(x^t) \cdot \mathbf{Emb}_{true}(x^t)}{\|\mathbf{Emb}_{pred}(x^t)\| \cdot \|\mathbf{Emb}_{true}(x^t)\|} \qquad (4)$$

To obtain the final Anomaly Score $\mathcal{AS}$, where higher values indicate greater likelihood, we perform normalization and integration of Equation equation 1, Equation equation 2, Equation equation 3 and Equation equation 4 as follows,

$$\mathcal{AS}^t = \sum_{i=1}^{4} w_i(1 - s_i) \qquad (5)$$

where $s_1 = LM/LM_m$, $s_2 = B_4$, $s_3 = F_1$, and $s_4 = ES$. Here, $LM_m$ denotes the upper bound used to normalize the margin score, which is set to 5.0 in our experiments. The weight $w_i$ corresponds to each evaluation metric.

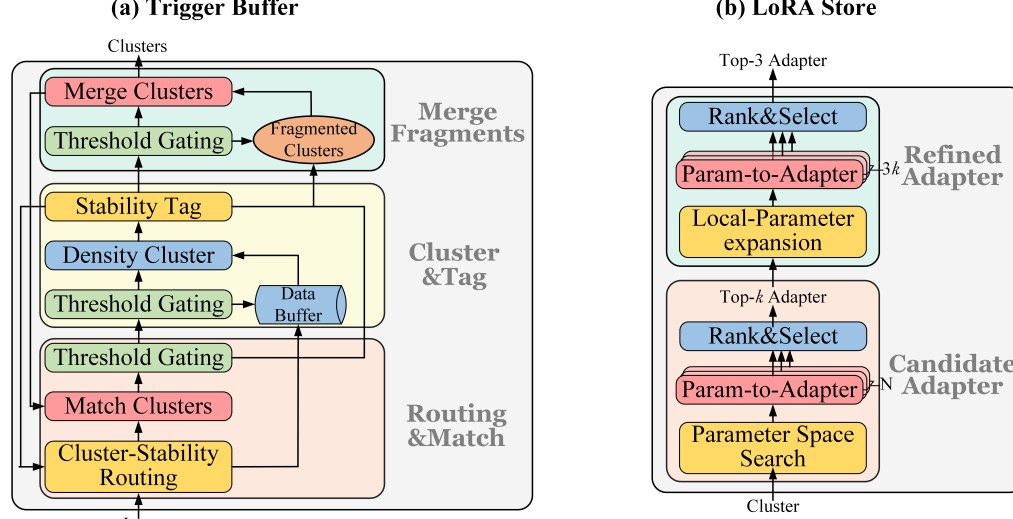

Figure 2: (left) Trigger Buffer consists of three layer: routing and match, cluster and tag, merge fragments. (right) LoRA Store consists of two layer: candidate adapter and refined adapter.

**Threshold Gating.** The input $x^t$ is processed by the Response Scoring layer to produce the predicted answer score $\mathcal{AS}^t$. A threshold $\tau$ is applied to define the gating decision function $g^t$ and $g^t = 1$ means that the inputs $x^t$ is identified as OOD data and submitted to the Trigger Buffer module. The formula is,

$$g^t = \begin{cases} 1, & \mathcal{AS}^t \geq \tau \\ 0, & \mathcal{AS}^t < \tau \end{cases} \tag{6}$$

### 3.2 TRIGGER BUFFER

The Trigger Buffer module in Figure 2 (left) clusters samples from Trigger module. Despite established clustering techniques, two challenges persist: ① **limited data**, due to dynamic triggering at test time; and ② **streaming data**, which arrives incrementally rather than in bulk, complicating clustering. To address the above challenges, we designed the Streaming Buffer Clustering (SBC) algorithm in Appendix A.1.

**Routing and Match.** Since data is provided externally, understanding the scope of knowledge domains often facilitates localization, similar to the way humans localize knowledge. As for Cluster-Stability Routing layer in Figure 2 (left), we conceptualize the data scope as a set of buckets. Therefore, for an input $x^t$, we first identify its corresponding bucket (i.e., data scope). We then retrieve the set of structural labels $\{s_1, s_2, \dots\}$:

$$s_i^t = 1, \; cluster \; i \; is \; stable \tag{7}$$

If all structural labels within the bucket are absent (i.e., equal to zero), the input $x^t$ is temporarily stored in the Data Buffer of the bucket. Otherwise, the inputs $x^t$ are match existing clusters using the cluster center $\mathcal{C}_c^t$, keywords of cluster $k_{\mathcal{C}^t}$ and keywords of inputs $k_{x^t}$ as follows,

$$Match(\mathcal{C}_c^t, x^t, k_{x^t}, k_{\mathcal{C}^t}) = \alpha \frac{\mathbf{Emb}(x^t) \cdot \mathbf{Emb}(\mathcal{C}_c^t)}{\|\mathbf{Emb}(x^t)\| \cdot \|\mathbf{Emb}(\mathcal{C}_c^*))\|} + (1 - \alpha) \frac{|k_{x^t} \cap k_{\mathcal{C}^t}|}{max \, |k_{\mathcal{C}^*}|} \tag{8}$$

where the weights are $\alpha = 0.7$ in the experiment. The threshold gating layer in Figure 2 (left) is applied again to evaluate the Match score. The inputs $x^t$ that exceed the threshold are added to the

selected clusters $\mathcal{C}^t$. If $x^t$ does not find a suitable cluster, a new cluster is created for it, potentially forming a Fragmented Cluster.

**Cluster and Tag.** Check whether the amount of data in the bucket corresponding to input $x^t$ has reached the threshold set by the threshold gating layer. If the threshold is not met, the clustering process for this input $x^t$ terminates; otherwise, Density Cluster in Figure 2 (left) is triggered. The SBC algorithm applies HDBSCAN (Campello et al., 2013) to density clustering, with two main advantages: ① it adapts to dynamic and unknown cluster structures in streaming data, and ② its density-based approach is robust to small, nonuniform datasets, effectively identifying high-density regions. Because data arrive as a stream, HDBSCAN alone does not necessarily produce stable clusters, which may impair the effectiveness of subsequent LoRA fine-tuning. To this end, we propose two metrics to quantify clustering stability:

1. Adjusted Random Index (ARI), measuring label consistency across successive clusterings. $RI$ is used to represent the Rand Index, and $\mathbb{E}[RI]$ is the expected value under random partitioning. It is defined as,

$$ARI = \frac{RI - \mathbb{E}[RI]}{max(RI) - \mathbb{E}[RI]} \tag{9}$$

2. Average Cosine Similarity $Emb_C$, quantifying semantic coherence between new and previous cluster centroids. $\mathcal{C}_c^{t-m}$ represents the cluster center at time $t-i$ before the cluster change, and $\mathcal{C}_c^t$ represents the clustering at time t. The formula is,

$$Emb_C(\mathcal{C}_c^{t-m}, \mathcal{C}_c^t) = \frac{\mathbf{Emb}(\mathcal{C}_c^{t-m}) \cdot \mathbf{Emb}(\mathcal{C}_c^t)}{\|\mathbf{Emb}(\mathcal{C}_c^{t-m})\| \cdot \|\mathbf{Emb}(\mathcal{C}_c^t)\|} \tag{10}$$

After obtaining the scores for $ARI$ and $Emb_C$, we set threshold values $\tau_{ARI}$ and $\tau_{Emb}$ to determine whether a cluster is labeled as stable and set $s_i^t = 1$. The formula is as follows.

$$ARI > \tau_{ARI} \ \cup \ Emb_C(\mathcal{C}_c^{t-m}, \mathcal{C}_c^t) > \tau_{Emb} \ \rightarrow \ s_i^t = 1 \tag{11}$$

At this point, if the stable label $s_i^t$ of a cluster changes, we update it as part of the Cluster-Stability Routing for the next input $x^{t+1}$. If a cluster maintains a stable state for an extended period but contains only a small and unchanging number of data, fragmented clusters may emerge.

**Merge Fragments.** Given that streaming inputs can form fragmented clusters, the threshold gating layer is applied to assess the number of stable tags $s_{x^t}$ within such clusters. If the number of stable tags exceeds the threshold, the Merge Clusters in Figure 2 (left) is triggered. At this point, we determine whether cluster merging is necessary by calculating the embedding similarity between cluster centers $\mathcal{C}_i^t$ and the *stable* label. The formula is,

$$\frac{\mathcal{C}_i^t \cdot \mathcal{C}_j^t}{\|\mathcal{C}_i^t\| \|\mathcal{C}_j^t\|} > \tau, \ cluster \ i \ and \ cluster \ j \ can \ be \ merged \tag{12}$$

If there are no clusters to be merged, the clustering process for this input $x^t$ will terminate; if clusters need to be merged, the new cluster's keywords, cluster centers, and other information must be updated in sync so that the Match Cluster can handle the subsequent input $x^{t+1}$.

## 3.3 LoRA Store

The LoRA storage module is inspired by how the human brain consolidates long-term memories during sleep. Therefore, we propose integrating fine-tuning with the endogenous memory of LLMs, forming a parameterized adapter that can internalize specific knowledge. Specifically, the LoRA Store module is responsible for fine-tuning the stable clusters in the Trigger Buffer at fixed intervals. Two major challenges remain: ① different types of datasets require specific tuning to obtain optimal configurations; and ② streaming inputs may alter the optimal adapter over time, necessitating the storage of multiple adapters to handle dynamic changes. To efficiently fine-tune parameters with limited data and diverse atomic tasks, we propose a CLO algorithm(see Appendix A.2) and the overall process is illustrated in Figure 2(right).

**Candidate Adapter**    After a fixed time interval $T$, we fine-tune the cluster $\mathcal{C}_s$ with the stable label. Because this is automated tuning, we let $\theta_{r,i}$ denote the $i$-th candidate value for the rank parameter in LoRA fine-tuning, and let $\theta_{l,j}$ denote the $j$-th candidate value for the learning rate in LoRA fine-tuning. In addition, epoch, LoRA alpha, dropout, batch size, and target modules are also included as hyperparameters. Each hyperparameter is assigned multiple candidate values, forming the hyperparameter space $\Theta$,

$$\Theta = \{\theta_{r,1}, \theta_{r,2}, \dots, \theta_{l,1}, \theta_{l,2}, \dots\} \tag{13}$$

By selecting one candidate value for each hyperparameter, we construct $N$ initial configuration sets, denoted as,

$$\mathcal{P}n = \{\theta_{r,i}, \theta_{l,j}, \dots\}, \ n = \{1, \dots, N\} \tag{14}$$

Based on the parameters of these $N$ initial configuration sets, we fine-tune the cluster $\mathcal{C}_s$ and evaluate the cross-entropy loss $CE$ and accuracy $Acc$ of each fine-tuned adapter $y_n^T$, recording them as the adapter's evaluation metrics. Finally, the fine-tuned adapters obtained from these $N$ initial configuration sets are sorted first by accuracy (descending), and then by cross-entropy loss (ascending), with the top-$k$ adapters selected. The formula is,

$$arg \ Top_k^{\preceq lex}\{Acc(y_n^T), -CE(y_n^T)\}_{n=1}^N \tag{15}$$

**Refined Adapter**    Based on the top-$k$ configurations obtained by the Candidate Adapter module, we first parse the logs to obtain the corresponding configuration sets $\mathcal{P}(i)i = 1^k$. We perform a nearest-neighbor search in the parameter space of rank and learning rate around the top-$k$ configurations, as these two hyperparameters have the greatest impact on fine-tuning across different inference settings (see Figure 18 and Appendix C.3). The nearest-neighbor parameter space for these two hyperparameters is defined as follows,

$$\mathcal{N}_{k_i} = \{\theta_{r,k_i} - 2, \theta_{r,k_i}, \theta_{r,k_i} + 2\} \times \{0.7 * \theta_{l,k_i}, 0.8 * \theta_{l,k_i}, 0.9 * \theta_{l,k_i}\} \tag{16}$$

For the $i$-th adapter among the top-$k$, we select 3 nearest-neighbor configuration sets, which yields $3k$ nearest-neighbor configuration sets in total. We rank the adapters trained from these $3k$ configuration sets using the criterion, and obtain the top-3 adapters. The formula is,

$$arg \ Top_3^{\preceq lex}\{Acc(y_n^T), -CE(y_n^T)\}_{n=1}^{3k} \tag{17}$$

The adapter configurations and weight parameters are stored locally for subsequent use in similar queries. That is, when new input $x^{t+m}$ arrives in Figure refframework, the top-3 fine-tuned adapter can be triggered and call it to answer the question.

## 4    EXPERIMENTS

### 4.1    EXPERIMENTAL SETTINGS

All experiments are conducted on a single NVIDIA A100 40GB GPU using LLaMA-2-7B as the base model. Parameter-efficient fine-tuning is performed via LoRA across all settings. For trigger detection and clustering, we employ BGE-large-en-v1.5 as the embedding model. Evaluation is conducted on four datasets designed to capture both in-distribution (ID) and out-of-distribution (OOD) behaviors. We use TriviaQA as the ID benchmark, which matches the model's pretraining distribution. To evaluate generalization under knowledge gaps, we include three OOD datasets from distinct domains: PubMedQA (biomedical Q&A), LexGlue (legal summarization), and GSM8K (mathematical reasoning). These datasets are chosen to represent structurally diverse, low-coverage regions unlikely to be encountered during pretraining.

### 4.2    MODULE-WISE EVALUATION OF SAGE

**Evaluation of Trigger Detection Accuracy.**    To fairly evaluate the detection capability of the Trigger module, we constructed a mixed test set containing equal proportions of ID and OOD samples.

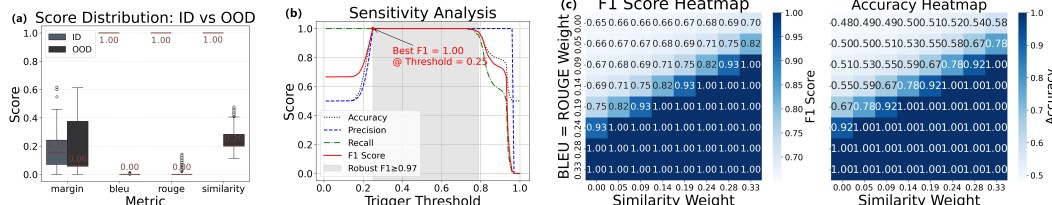

Figure 3: Parameter analysis of the Trigger Module. (a) False positive rates on ID vs. OOD samples. Red lines in boxplots indicate the mean; the logits margin metric is normalized. (b) Threshold sensitivity analysis; the gray area marks the optimal threshold range. (c) Heatmap of score weight sensitivity. The logits margin weight is computed dynamically from BLEU, ROUGE-L, and embedding similarity differences.

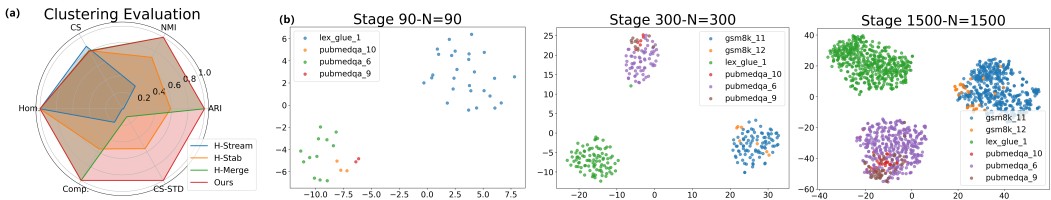

Figure 4: Results of the Trigger Buffer Module evaluation. (a) Evaluation of Streaming HDBSCAN Variants on Clustering Metrics: Homogeneity, ARI, NMI, Completeness, Central Similarity (CS), and Cluster Sample Std. (CS-STD). (b) Dynamic t-SNE Visualization as Data Volume (N) Increases.

The OOD samples were drawn in equal parts from three domain-specific datasets: PubMedQA, LexGlue, and GSM8K. All samples were uniformly converted into a structure consisting of the fields "*question*", "*full prompt*", "*real-answer*", and "*label*", where *label = 0* denotes ID samples and *label = 1* denotes OOD samples.

To evaluate the Trigger module's accuracy and robustness, we conducted three experiments (Figure 3). Figure 3(a) shows that semantic indicators (BLEU, ROUGE, similarity) score near 1 on ID samples and significantly lower on OOD, indicating strong discriminative power. Logits margin shows smaller variance but clear mean differences. Figure 3(b) demonstrates stable performance across a broad threshold range (0.22–0.79), confirming robustness. Figure 3(c) highlights that increasing weights for BLEU and ROUGE-L improves accuracy, emphasizing the role of semantic cues. Overall, **the Trigger module reliably separates ID/OOD samples with stable performance under varying thresholds and indicator weights**. In subsequent experiments, we fix the threshold at 0.5 with equal weights and observe 100% ID/OOD separation (see Appendix B.1).

**Clustering Analysis of Triggered Buffer.** Given the Trigger module's effectiveness, we assume no ID samples entering the Trigger Buffer. Thus, we evaluate its clustering performance using only OOD samples, drawn from the same dataset used in Trigger evaluation.

To evaluate the clustering capability of the Trigger Buffer, we conduct three experiments shown in Figure 4 and Figure 5. Figure 4(a) compares six clustering metrics across different strategies. Our SBC algorithm, which combines HDBSCAN, a stability check, and a merge mechanism, achieves the best overall performance. While all methods show good homogeneity due to bucket-based grouping, HDBSCAN alone yields unstable clusters, reflected in low ARI and CS-STD. Adding the stability check improves consistency, and the merge strategy reduces intra-cluster dispersion. SBC algorithm integrates both for consistent and compact clustering. Figure 4(b) shows clusters

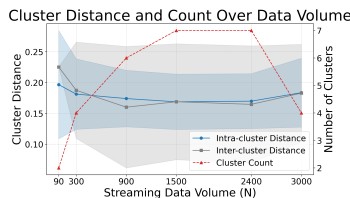

Figure 5: Cluster distance and cluster count over streaming data volume.

becoming more compact as data accumulates. Figure 5 tracks three metrics: decreasing intra-cluster distance and cluster count indicate redundancy reduction, while stable inter-cluster distance con-

firms structural robustness. Overall, **the Trigger Buffer produces compact, stable clusters with convergence behavior in streaming OOD settings.**

Table 1: Evaluating the Accuracy of the LoRA Store Module on Diverse Datasets: Highlighting the Role of Atomic Tasks. The subscript numbers in the table denote the size of the training set.

| Method | | PubMedQA(%) | | | | LexGlue(%) | | | | GSM8K(%) | | | |
|---|---|---|---|---|---|---|---|---|---|---|---|---|---|
| | | Acc | Macro Avg | | | Acc | Macro Avg | | | EM | MAE | MSE | NER |
| | | | P | R | F1 | | P | R | F1 | | | | |
| Base Model | | 50.60 | 75.15 | 50.60 | 34.65 | 12.53 | 17.75 | 12.53 | 13.19 | 0 | 175 | 32337 | - |
| $LoRA_{80000}$ | | 67.60 | 75.00 | 67.59 | 65.01 | 65.43 | 73.99 | 85.22 | 65.43 | 20.00 | 481 | $1.8 \times 10^6$ | **100** |
| $CA\text{-}LoRA_{30000}$ | | **84.80** | **85.71** | **84.80** | **84.70** | **72.21** | **79.63** | **72.21** | **75.50** | 23.38 | 92 | $3.4 \times 10^8$ | **100** |
| $LStore_{500}$ (random datasets) | $Adp_1$ | 43.2 | 48.36 | 43.62 | 34.07 | 34.15 | 27.62 | 34.50 | 34.15 | 6.44 | 11609 | $3.6 \times 10^{10}$ | 99.17 |
| | $Adp_2$ | 41.4 | 50.22 | 50.84 | 34.23 | 40.08 | 47.79 | 40.08 | 37.10 | 8.89 | 11594 | $3.6 \times 10^{10}$ | 100 |
| | $Adp_3$ | 58.15 | 50.76 | 51.52 | 50.17 | 36.08 | 38.12 | 35.98 | 36.08 | 6.72 | 14995 | $3.9 \times 10^{10}$ | 99.17 |
| $LStore_{500}$ (atomic datasets) | $Adp_1$ | 79.48 | 100 | 69.20 | 79.47 | 65.00 | 64.79 | 65.00 | 64.73 | **92.32** | **95** | $8.3 \times 10^5$ | **93.99** |
| | $Adp_2$ | 77.78 | 90.05 | 69.36 | 77.78 | 69.44 | 81.45 | 69.44 | 66.95 | **81.27** | 473 | $3.0 \times 10^7$ | 63.13 |
| | $Adp_3$ | 90.69 | 98.74 | 84.00 | 90.60 | 57.14 | 60.61 | 57.14 | 53.33 | **72.13** | 13917 | $4.7 \times 10^9$ | 57.52 |

**Dynamic Adaptation of LoRA Store.** To assess the necessity of low-rank fine-tuning for atomic tasks, we evaluate the LoRA Store module (Table 1). Comparisons are made against three baselines: the Base Model (Llama2-7B), standard LoRA (fine-tuned on the full dataset), and CA-LoRA (fine-tuned after static clustering). Results in Table 1 demonstrate the effectiveness of our dynamic adaptation approach in integrating new knowledge. All baseline models used the mixed-task datasets provided. PubMedQA's and LexGlue tasks enabled strong baseline performance, with CA-LoRA outperforming direct LoRA fine-tuning, highlighting clustering's benefit. In contrast, GSM8K's complex arithmetic tasks posed greater challenges, reflected by low exact match (EM) scores despite reasonable numeric extraction (NER). Simple clustering and training yielded limited gains. During LoRA Store training, PubMedQA and LexGlue showed minimal improvement, likely due to task heterogeneity and difficulty in atomic decomposition. However, **decomposing GSM8K's level-1 subset (tasks with 1–2 computation steps) significantly boosted reasoning accuracy, validating the effectiveness of atomic tasks**.

To evaluate the LoRA Store's effectiveness, heatmaps (Figure 18) were created. Accuracy can vary significantly (up to 80%) and loss decreases depending on the rank and learning rate. Optimal configurations vary by rank, e.g., $r = 6$ at lr $= 7.92 \times 10^{-5}$, and $r = 10$ at lr $= 1.99 \times 10^{-4}$. These results highlight **the necessity of exploitation optimization, as performance varies across rank and learning rate.**

Figure 6: Heatmap of accuracy and cross-entropy loss for LoRA parameter: examining the necessity of exploitation optimization.

### 4.3 END-TO-END EVALUATION OF SAGE

We evaluated SAGE by integrating all three modules, focusing on the GSM8K Level-1 dataset due to LoRA Store's superior performance on atomic tasks. Three baselines were used: Base Model, LoRA, and CA-LoRA. Final results, including LoRA Store fine-tuning, are shown in Table 7. Results show that SAGE with dynamic clustering outperforms the unclusterd LoRA Store. We further performed the Wilcoxon signed-rank test to assess statistical significance. **The result ($W = 0.0, p = 0.0039$) confirms that the improvement of**

Figure 7: Evaluating the Accuracy of the SAGE. The ↑ indicates the maximum value, and the ↓ indicates the minimum value.

| Method | | GSM8K(%) | | | |
|---|---|---|---|---|---|
| | | EM | MAE | MSE | NER |
| Base Model | | 2.22 | 2258 | $2.0 \times 10^8$ | - |
| $LoRA_{499}$ | | 7.44 | 2079 | $2.7 \times 10^8$ | 99.79 |
| $CA\text{-}LoRA_{484}$ | | 10.64 | 1096 | $8.1 \times 10^7$ | 99.80 |
| $LStore_{499}$ (ours) | $Adp_1$ | 92.32 | 95 | $8.3 \times 10^5$ | 93.99 |
| | $Adp_2$ | 81.27 | 473 | $3.0 \times 10^7$ | 63.13 |
| | $Adp_3$ | 72.13 | 13917 | $4.7 \times 10^9$ | 57.52 |
| $SAGE_{315}$ (ours) | $Adp_1$ | **84.95** | 28185 | $1.5 \times 10^7$ | 78.58 |
| | $Adp_2$ | **99.80** | 0.05 | 1.36 | 99.40 |
| | $Adp_3$ | **99.80**↑ | **0.01**↓ | **0.16**↓ | **100**↑ |
| **Random Seed of EM** | | | | $97.16_{\pm 4.65}$ | |

**SAGE over the baseline is statistically significant ($p < 0.01$).** Clustering refines the dataset, reduces noise, and improves convergence with fewer samples. During evaluation, the Top-3 adapters were retained and the effectiveness of this approach was verified. While all reached 100% training accuracy, only Adapter$_3$ Adp$_3$ generalized well, achieving the best test EM, MSE, MAE, and digit extraction rate. To assess stability, we shuffled the data with different seeds. As shown in Appendix B.2, **SAGE's performance remained consistent, indicating strong robustness to data order**. Also, the shaded rows in Table 7 report the mean and variance of EM scores over 9 adapters from the random seed experiments.

## 4.4 ABLATION STUDY

We omit ablation studies on the Trigger and LoRA Store modules, as their functionalities have been validated in prior experiments: the Trigger module reliably distinguishes ID and OOD data to control dynamic inference, while the LoRA Store handles dynamic hyperparameter tuning, outperforming static baselines. Our ablation focuses on the Trigger Buffer and atomic task datasets. As shown in Table 8, we compare following baseline settings: (1) LoRA Store without task decomposition or clustering, (2) with task decomposition but no Trigger Buffer, and (3) with static clustering (HLStore, KLStore). **SAGE outperforms all variants in EM, MAE, and NER, confirming the effectiveness of atomic task splitting and dynamic clustering via the Trigger Buffer.**

Figure 8: Ablation Study of SAGE on the GSM8K Dataset.

| Method | | GSM8K(%) | | | |
|---|---|---|---|---|---|
| | | EM | MAE | MSE | NER |
| LStore$_{499}$ (random) | Adp$_1$ | 6.44 | 11609 | $3.6 \times 10^{10}$ | 99.17 |
| | Adp$_2$ | 8.89 | 11594 | $3.6 \times 10^{10}$ | 100 |
| | Adp$_3$ | 6.72 | 14995 | $3.9 \times 10^{10}$ | 99.17 |
| LStore$_{499}$ (level-1) | Adp$_1$ | 92.32 | 95 | $8.3 \times 10^5$ | 93.99 |
| | Adp$_2$ | 81.27 | 473 | $3.0 \times 10^7$ | 63.13 |
| | Adp$_3$ | 72.13 | 13917 | $4.7 \times 10^9$ | 57.52 |
| HLStore$_{484}$ (HDBSCAN) | Adp$_1$ | 89.77 | 309 | $1.9 \times 10^7$ | 98.97 |
| | Adp$_2$ | 78.70 | 21812 | $9.6 \times 10^9$ | 89.26 |
| | Adp$_3$ | 96.69 | 429 | $8.9 \times 10^7$ | 100 |
| KLStore$_{329}$ (K-means) | Adp$_1$ | 97.55 | 14 | 31717 | 99.09 |
| | Adp$_2$ | 74.47 | 377 | $6.5 \times 10^5$ | 100 |
| | Adp$_3$ | 92.40 | 646 | $1.3 \times 10^8$ | 100 |
| SAGE$_{315}$ (ours) | Adp$_1$ | 84.95 | 28185 | $1.5 \times 10^7$ | 78.58 |
| | Adp$_2$ | **99.80** | **0.05** | **1.36** | **99.40** |
| | Adp$_3$ | **99.80**↑ | **0.01**↓ | **0.16**↓ | **100**↑ |

## 5 RELATED WORK

**Self-Adaptive Mechanisms in LLMs** Self-adaptive mechanisms are considered essential for memory-augmented LLMs. Early work identified key-value-like storage behavior in Transformer FFNs (Geva et al., 2021), inspiring enhanced memory routing (Wu et al., 2022; Berges et al., 2024; Li et al., 2025), long-context inputs (Wang et al., 2023), and retrieval-augmented pathways (Borgeaud et al., 2022) to improve knowledge persistence. Memory[3](Lewis et al., 2020; Liu et al., 2024) spurred interest in retrieval-augmented generation for knowledge updating. More recently, MemoryLLM (Wang et al., 2024) shifted focus to self-adaptive mechanisms (Zhong et al., 2022).

**LoRA-Based Adaptation and Reusability** LoRA have advanced parameter-efficient fine-tuning (He et al.). Increasing focus has been placed on LoRA's modularity and reusability (Ostapenko et al., 2024; Xu et al., 2024; Li et al., 2024; Valipour et al., 2023; Liao et al., 2025), with methods like Switch-LoRA (Kong et al., 2024) supporting dynamic adapter selection and plug-and-play use. Optimizing LoRA training efficiency, including hyperparameter tuning and update latency, remains an active research direction.

## 6 CONCLUSION AND DISCUSSION

We presented the SAGE framework, which decomposes reasoning into atomic subtasks and dynamically fine-tunes LoRA parameters via failure-triggered detection and clustering. This approach enables adaptive test-time updates, improving the model's ability to handle OOD knowledge. While the current design favors simplicity, its strong empirical performance suggests a promising foundation for future refinement. Further research on SAGE could accelerate the development of endogenous memory in LLMs, for example by replacing the trigger module with neural architectures to enhance flexibility and scalability. Moreover, although the atomic-task approach performs well on simpler problems, extending it to support complex multi-step reasoning remains an important avenue for future work. More broadly, SAGE demonstrates how combining detection, clustering, and efficient adaptation can provide a principled route toward adaptive reasoning in large language models.

## 7 REPRODUCIBILITY STATEMENT

The code for our proposed SAGE method, along with the scripts used for experimental evaluation, is included in the supplementary materials and can be directly extracted. In addition, the code for constructing the atomized datasets, as well as the atomized datasets themselves, are also provided in the supplementary materials (availabel at https://anonymous.4open.science/r/SAGE-C7F3). The detailed methodology for atomized dataset construction is described in Appendix C.3.

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

# APPENDIX

## A IMPLEMENTATION DETAILS

### A.1 DETAILS FOR TRIGGER BUFFER MODULE OF SBC ALGORITHM

The Trigger Buffer module of Streaming Buffer Clustering (SBC) algorithm details are as follows:

For each newly arrived abnormal sample $x$, the SBC algorithm infers its structure tag and extracts its semantic embedding vector and keyword set for subsequent similarity matching. If the cluster for the structure tag $s$ is deemed "*stable*" (i.e., a structure cluster $\mathcal{C}_s$ already exists), SBC algorithm computes the similarity between the current sample and all candidate clusters under the same tag, selecting the one with the highest score (Line 5). The clustering score $\gamma$ is a weighted combination of semantic embedding similarity and keyword overlap. If $\gamma$ exceeds the threshold $\tau$, the sample $x$ is added to the selected cluster $\mathcal{C}^*$ (Line 8), and the assignment result is returned, terminating the process. If the structure cluster is not stable or the score does not meet the threshold, the sample is temporarily stored in the buffer $\mathcal{B}_s$ for tag $s$ (Line 12), awaiting further processing by the delayed clustering mechanism.

When a newly abnormal sample $x$ causes the number of samples in the buffer associated with structure tag $s$ to reach the predefined threshold $T$ (Line 14), the SBC algorithm triggers batch clustering. The system extracts the semantic embedding vectors $\mathbf{e}_i$ of all buffered samples (Line 15), then applies HDBSCANfor unsupervised density-based clustering. However, HDBSCAN alone cannot guarantee clustering *stability*. To address this, we introduce two stability check metrics: ① the **Adjusted Rand Index (ARI)**, measuring label consistency across successive clusterings; and ② the **Average Cosine Similarity**, quantifying semantic coherence between new and previous cluster centroids. When both exceed their thresholds $\eta_{\text{ARI}}$ and $\eta_{\cos}$, clustering is considered stable. The buffered samples are migrated to a newly formed formal cluster $\mathcal{C}_{\text{new}}$, the buffer $\mathcal{B}_s$ cleared (Line 21), and the structure tag $s$ marked as "*stable*" (Line 22).

Given that streaming input can fragment clusters, when the number of stable clusters for tag $s$ exceeds the threshold, an **inter-cluster merge** (Line 24) is triggered. Clusters are merged if their centroids' cosine similarity exceeds threshold $\delta$, reducing redundancy and increasing sample density. If batch clustering fails to meet stability criteria, the result is discarded and samples remain buffered for future clustering(Line 28). Finally, samples unassigned to any cluster and without triggered clustering are marked as "*unassigned*" until clustering conditions are met (Line 31).

---

**Algorithm 1: Streaming Buffer Clustering (SBC)**

---

**Input:** Incoming anomaly sample $x$, data buffer $\mathcal{D}$, current clusters $\mathcal{C}$, structure tag $s$
**Output:** Cluster assignment or data buffer update

1: $s \leftarrow \text{InferStructure}(x)$
2: $e \leftarrow \text{GetEmbedding}(x)$
3: $k \leftarrow \text{ExtractKeywords}(x)$
4: **if** IsClusterStable(s) == True **then**
5:     $(\mathcal{C}^*, \gamma) \leftarrow \text{FindBestCluster}(s, e, k)$
6: # $\gamma$: weighted score combining embedding similarity $e$ and keyword overlap $k$
7:     **if** $\gamma \geq \tau$ **then**                             ▷ Exceeding threshold $\tau$
8:        Assign $x$ to cluster $\mathcal{C}^*$
9:        **return** cluster assignment
10:     **end if**
11: **else**
12:     $\text{AddToBuffer}(x, e, k, \mathcal{D}_s)$
13: **end if**
14: **if** $|\mathcal{D}_s| \geq T$ **then**                            ▷ Exceeding clustering threshold $T$
15:     $e \leftarrow \text{ExtractEmbeddings}(\mathcal{D}_s)$
16:     $h \leftarrow \text{HDBSCAN}(e)$
17:                                  ▷ Density clustering without preset count
18:     $(\text{ARI}, \text{Sim}) \leftarrow \text{EvalStability}(h, h^{(t-1)}, e, e^{(t-1)})$
19:     **if** $\text{ARI} \geq \theta_{\text{ari}} \vee \text{Sim} \geq \theta_{\text{sim}}$ **then**
20:        $\text{CreateNewClusters}(h, \mathcal{D}_s)$
21:        $\mathcal{D}_s \leftarrow \emptyset$                             ▷ Clear buffer
22:        $\text{StableFlag}_s \leftarrow \text{True}$
23:        **if** $|\mathcal{C}_s| \geq 3$ **then**
24:           $\text{MergeSimilarClusters}(\mathcal{C}_s)$
25:                               ▷ Merge clusters with centroid similarity
26:        **end if**
27:     **else**
28:        **return**                            ▷ Clustering unstable, wait more
29:     **end if**
30: **end if**
31: **return** $x$ as "Unassigned"                      ▷ No matching cluster

---

## A.2 DETAILS FOR LoRA STORE MODULE OF CLO ALGORITHM

The LoRA Store module of Cluster-Aware LoRA Optimization (CLO) algorithm, a clustering-based method that automatically searches for the optimal LoRA configuration $\theta^*$, saving the corresponding adapter for subsequent inference or combination, details are as follows:

---

**Algorithm 2:** Cluster-Aware LoRA Optimization (CLO)

---

**Input:** Incoming stable cluster $\mathcal{C}_s$, base model $\mathcal{M}$, tokenizer $\mathcal{T}$, parameter space $\Theta$ , top-$k$ configs $k$

**Output:** Optimized LoRA config $\theta^*$ and LoRA adapter $\mathcal{A}$

```
 1: 𝒫_n ← InitialConfigs(Θ, n)                        ▷ Initialize n seed configs
 2: ℛ_0 ← []                                          ▷ Initial training results
 3: for all θ ∈ 𝒫_n do
 4:      r ← TrainLoRA(θ, 𝒞, ℳ, 𝒯)
 5:      ℛ_0 ← ℛ_0 ∪ {r}
 6: end for
 7: ℛ_0 ← SortByScore(ℛ_0)
 8: # maximize accuracy, break ties by minimizing loss
 9: 𝒫_top ← Top-KConfigs(ℛ_0, k)
10: ℛ_1 ← []                                          ▷ Refined training results
11: for all θ_top ∈ 𝒫_top do
12:      𝒫_opt ← LocalParamSearch(θ_top)
13: #local search over rank and learning rate near θ_top
14:      for all θ′ ∈ 𝒫_opt do
15:          r′ ← TrainLoRA(θ′, 𝒞, ℳ, 𝒯)
16:          ℛ_1 ← ℛ_1 ∪ {r′}
17:      end for
18: end for
19: ℛ_all ← ℛ_0 ∪ ℛ_1
20: ℛ_all ← SortByScore(ℛ_all)
21: 𝒜_LoRA ← Top-KResults(ℛ_all, 3)
22: SaveBuffer(𝒜_LoRA)
23: return 𝒜_LoRA and θ^*
```

---

Given a cluster $\mathcal{C}$ with stable structure, the CLO algorithm randomly samples candidate configurations $n$ from the hyperparameter search space $\Theta$ to form the initial configuration set $\Theta_0$. Simultaneously, the system creates an empty list to store the results of each configuration after training, including validation accuracy, cross-entropy loss, save path, and corresponding LoRA adapter configuration information. The CLO algorithm then performs complete LoRA fine-tuning on each configuration in $\mathcal{P}_n$ for the samples in the cluster $\mathcal{C}_s$ (Line 4), recording the training results to form a preliminary result set $\mathcal{R}_0$ (Line 5). Next, CLO algorithm ranks the results in $\mathcal{R}_0$ based on a sorting strategy that prioritizes validation accuracy firstly and cross-entropy loss secondly, selecting the Top-$k$ configurations (Line 9). This strategy accounts for the limited data size, prioritizing accuracy to ensure SAGE's correct response to similar inputs.

Based on these $k$ optimal configurations, CLO algorithm conducts exploitation optimization in their neighboring parameter spaces (Line 12), mainly adjusting the LoRA rank and learning rate. For each configuration in the fine-tuned set $\mathcal{P}_{opt}$, CLO algorithm continues LoRA fine-tuning, resulting in a refined result set $\mathcal{R}_1$ (Line 14). Finally, CLO algorithm merges the preliminary result set $\mathcal{R}_0$ with the refined set $\mathcal{R}_1$ into the total result set $\mathcal{R}\text{all}$ (Line 19), sorts it again by accuracy and cross-entropy loss, and selects the final Top-3 configurations (Line 21). The corresponding LoRA adapters are saved as the final optimized results $\mathcal{A}_{LoRA}$ (Line 22), and the three optimal adapters, along with the best configuration $\theta^*$, are returned for subsequent reasoning calls (Line 23). For new data with different characteristics, adaptation is achieved by triggering additional LoRA fine-tuning.

The CLO algorithm combining initial exploration with local refinement, making it suitable for atomic tasks characterized by sparse data and diverse atomic tasks. It ensures that each cluster has the optimal LoRA configuration for the current data, providing a solid foundation for subsequent adapter upgrades and dynamic fine-tuning.

# B RELATED WORK

## B.1 SELF-ADAPTING MECHANISMS IN LLMS

Since the emergence of LLMs, a series of studies have investigated memory mechanisms inspired by cognitive systems, particularly viewing LLMs as brain-like architectures. Early work observed that the feed-forward networks (FFNs) in Transformer models exhibit key-value-like storage behavior Geva et al. (2021), prompting efforts to enhance memory functionality by routing information through these components Wu et al. (2022); Berges et al. (2024); Li et al. (2025). This line of research also motivated the construction of long-context Wang et al. (2023) inputs and retrieval-enhanced pathways Borgeaud et al. (2022) to improve internal knowledge persistence. A significant milestone was marked by the introduction of Memory[3] Yang et al. (2024), which explicitly decomposed memory into three components: model parameters, explicit memory modules, and retrieved text. The study demonstrated that externalized memory not only facilitates faster knowledge acquisition but also serves as a viable mechanism for persistent knowledge representation in LLMs. This perspective helped establish the notion that memory accelerates learning and catalyzed a surge of interest in RAG Lewis et al. (2020); Liu et al. (2024) for knowledge updating and memory simulation. More recently, the development of MemoryLLM Wang et al. (2024) has shifted attention toward mechanisms for self-updating. Rather than solely focusing on constructing long-term or short-term memory banks, this work highlights the importance of enabling LLMs to selectively and autonomously revise their internal representations—marking Zhong et al. (2022) a conceptual shift from passive storage to active self-modification.

## B.2 LORA-BASED ADAPTATION AND REUSABILITY

In parallel with memory research, substantial progress has been made in advancing LoRA and its variants for parameter-efficient fine-tuning He et al.. Inspired in part by the success of sparse mixture-of-experts (MoE) Shazeer et al. (2017) architectures, recent studies have proposed sparsity-aware LoRA designs to support efficient and selective adaptation in data-limited settings Yeasar et al. (2025). Beyond sparsity, increasing attention has been given to the modularity and reusability Ostapenko et al. (2024); Xu et al. (2024); Li et al. (2024); Valipour et al. (2023); Liao et al. (2025) of LoRA modules. Approaches such as Switch-LoRA Kong et al. (2024) explore dynamic adapter selection and plug-and-play capabilities, enabling rapid composition and reuse across diverse tasks. Meanwhile, the training efficiency of LoRA—particularly in terms of hyperparameter optimization and update latency—has become an active area of research.

# C EXPERIMENTAL DETAILS

## C.1 DETAILS FOR TRIGGER MODULE

In further experiments, we fixed the threshold at 0.5 and set equal weights for all four indicators. We then evaluated the Trigger module independently under these test-time parameters. As shown in Figure 9, the module achieved 100% ID/OOD separation under this configuration, confirming that the chosen parameters fall within a favorable operational range. The OOD and ID score distributions in Figure 9(c) show that the OOD and ID samples fall within two distinct ranges, providing a sufficiently wide range of thresholds to distinguish the two classes. Together with the sensitivity analysis in Figure 3, these findings suggest that the Trigger module possesses not only high accuracy but also strong robustness and generalizability, making it well-suited to serve as a reliable activation mechanism for subsequent adaptive fine-tuning in LLM-based systems.

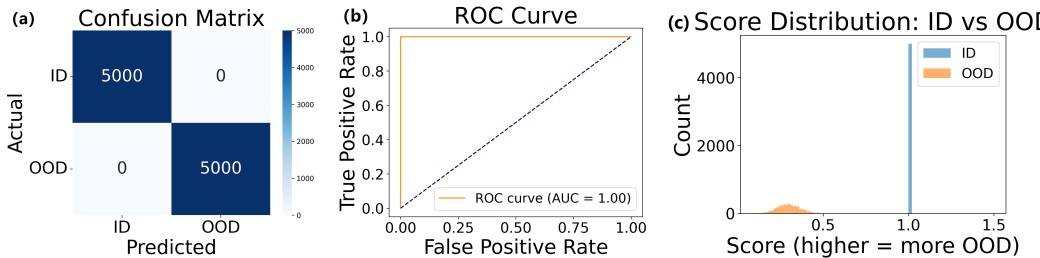

Figure 9: Results of the Trigger Module evaluation under formal test parameters. (a) Confusion matrix for ID and OOD samples. (b) ROC curve illustrating detection performance. (c)Anomaly score of ID datasets and OOD datasets.

To further validate the accuracy of the trigger model, we expanded the types of OOD samples. Unlike the short real answers (mainly from multiple-choice questions, typically "yes," "no," or numerical responses) used in the previous OOD evaluation, we additionally tested on text-based real answers (narrative responses, akin to short-answer questions). Specifically, we selected three OOD datasets: entailment [1], worldtree [2], and ARC [3]. As shown in Figure 10, the module achieved 100% ID/OOD separation under this setting, confirming that the chosen parameters lie within a favorable operational range.

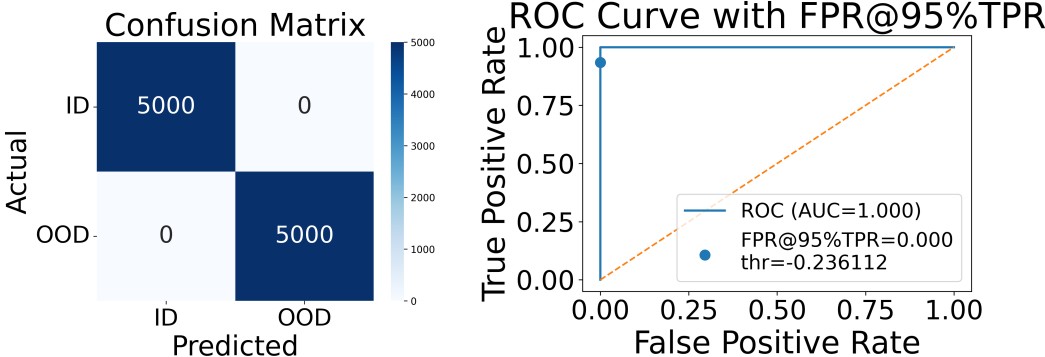

Figure 10: Results of the Trigger Module evaluation under TriveQA for ID and Entailment, Worldtree and ARC for OOD. (left) Confusion matrix for ID and OOD samples. (right) ROC curve illustrating detection performance.

Moreover, we conducted score and threshold sensitivity analyses on these datasets. The results, presented in Figure 11, demonstrate that even for datasets containing longer sentences. The OOD and ID score distributions also show that the OOD and ID samples fall within two distinct ranges, providing a sufficiently wide range of thresholds to distinguish the two classes. And the trigger module exhibits not only high accuracy but also strong robustness and generalizability.

---

[1] Available at https://huggingface.co/datasets/suzakuteam/entailment_bank

[2] Available at https://huggingface.co/datasets/nguyen-brat/worldtree

[3] Available at https://huggingface.co/datasets/KomeijiForce/ARC-Challenge-Explained-by-ChatGPT

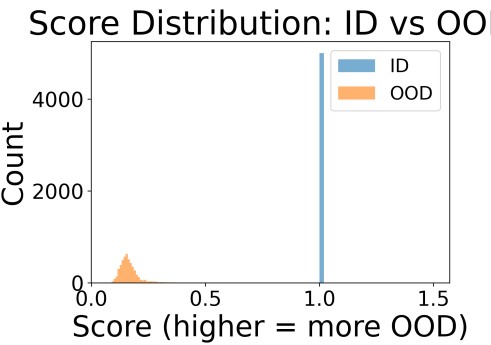 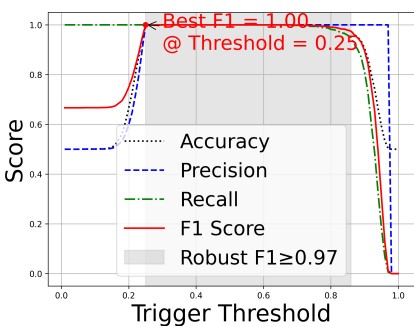

Figure 11: Results of the Trigger Module evaluation under TriveQA for ID and Entailment, Worldtree and ARC for OOD. (left) Anomaly score of ID datasets and OOD datasets. (right) Threshold sensitivity analysis; the gray area marks the optimal threshold range.

## C.2 DETAILS FOR TRIGGER BUFFER MODULE

To cope with streaming data, we designed multiple thresholds. Below is a sensitivity analysis of each threshold to verify the rationality of our SBC algorithm design.

Figure 12 shows the results of the similarity threshold in the threshold gating module, which comes after the match clusters module in the routing and matching process. Data values greater than this threshold are considered similar to a cluster and can be added to it. Since a lower DBI and a higher CH are desirable, and moderate values of unassigned rate and cluster number are preferred, the similarity threshold should be in the range of 0.55 to 0.65. This threshold is relatively sensitive and can serve as the primary factor for adjusting the clustering process.

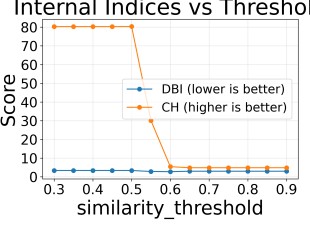 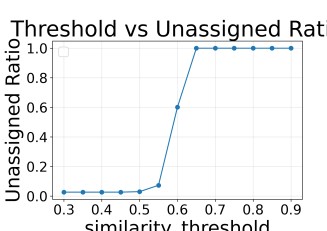 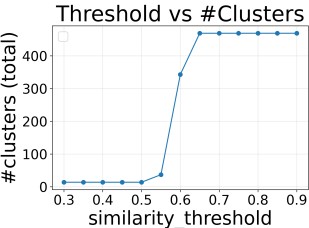

Figure 12: Similarity threshold sensitivity analysis for threshold gating after match clusters module. (left) Davies–Bouldin Index and Calinski–Harabasz Index for internal indices. (middle) Outlier Ratio for proper threshold. (right) Cluster Stability.

Figure 13 shows the results of the data buffer size threshold for the threshold gating module, which precedes the density clustering module in the cluster-and-tag process. When the threshold exceeds this value, the SBC algorithm triggers the density clustering module, and the HDBSCAN algorithm performs the clustering. The results show that a data buffer size between 5 and 40 yields the best performance. Adjusting this parameter can slightly improve clustering results, but the effect is not significant, indicating that it is suitable for secondary parameter tuning.

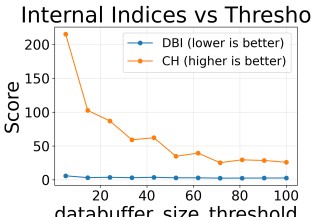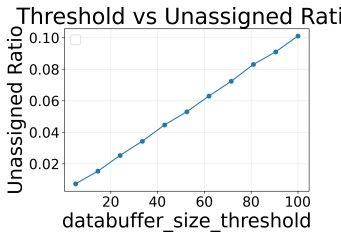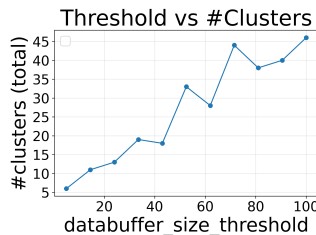

Figure 13: Data buffer size threshold sensitivity analysis for threshold gating before density cluster module. (left) Davies–Bouldin Index and Calinski–Harabasz Index for internal indices. (middle) Outlier Ratio for proper threshold. (right) Cluster Stability.

Figure 14 shows the results of the merge threshold for the threshold gating module, which precedes the merge clusters module in the merge fragments module. When the threshold exceeds this value, the SBC algorithm enters the fusion region to determine cluster centers. The results show that a threshold value greater than 6 is most effective, and the clustering performance remains largely unchanged. Therefore, this parameter can be used to enable or disable the merge operation, and adjustment is unnecessary unless the merge operation needs to be disabled.

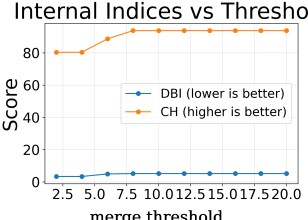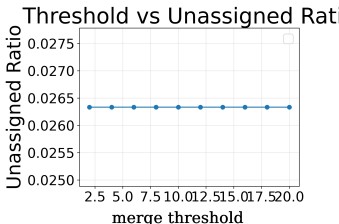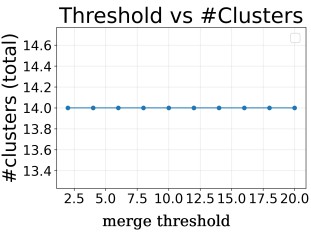

Figure 14: Merge threshold sensitivity analysis for threshold gating before merge clusters module.(left) Davies–Bouldin Index and Calinski–Harabasz Index for internal indices. (middle) Outlier Ratio for proper threshold. (right) Cluster Stability.

Figure 15 shows the results of the cluster center similarity threshold in the merge clusters module, which is part of the merge fragments process. When the threshold exceeds this value, the SBC algorithm executes the fragmented cluster fusion operation. The results show that setting the threshold between 0.8 and 1.0 yields the best performance, with clustering remaining largely stable across this range. Therefore, this parameter can be set as a fixed value within the algorithm and does not require further adjustment.

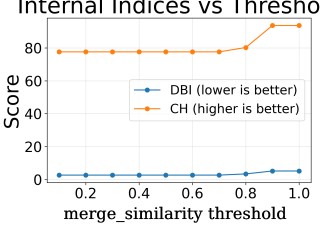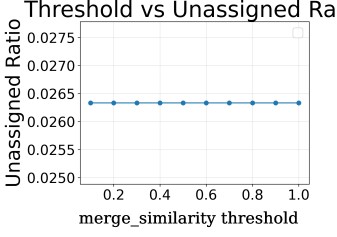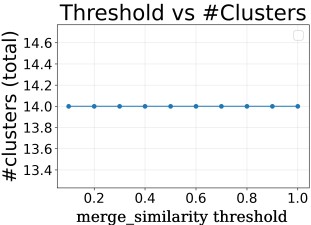

Figure 15: Cluster center similarity threshold sensitivity analysis for merge clusters module.(left) Davies–Bouldin Index and Calinski–Harabasz Index for internal indices. (middle) Outlier Ratio for proper threshold. (right) Cluster Stability.

Figure 16 shows the results of a sensitivity analysis of the Adjusted Random Index and Average Cosine Similarity thresholds. These thresholds are used in the density module of the cluster and tag process. The SBC algorithm requires two thresholds, and both must simultaneously exceed their respective values for the algorithm to assign a stability label to a cluster. This requirement also ensures

that the cluster can be matched with the data. Therefore, we conducted a detailed threshold analysis of these two parameters in a three-dimensional space. The results show that an Adjusted Random Index (ARI) threshold above 0.8 performs well. An Average Cosine Similarity (ACS) threshold between 0.5 and 0.9 also performs well. Both thresholds are very robust. Therefore, no adjustment of these threshold parameters is recommended.

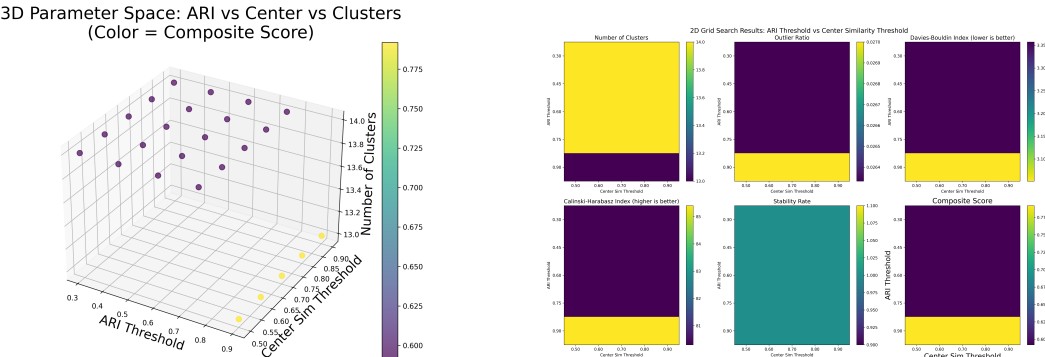

Figure 16: Ajusted Random Index and average Cosine Similarity thresholds sensitivity analysis for density cluster module. (left) 3d parameter for cluster stability of threshold. (right) 2d parameter results for cluster stability, outlier ratio, DBI, CH, stability rate and composite score.

Figure 15 shows the results of the match score weight in the match cluster module for routing and matching. This threshold determines the score weight parameter. Because there are only two weights that sum to 1, only one weight parameter was tested. The weight parameter shown in the figure is the coefficient of embedding similarity between the data and the cluster. The results show that setting this threshold between 0.5 and 0.7 performs best and clustering results are sensitive to changes in this weight. Therefore, this threshold should be prioritized as the primary factor for clustering adjustments.

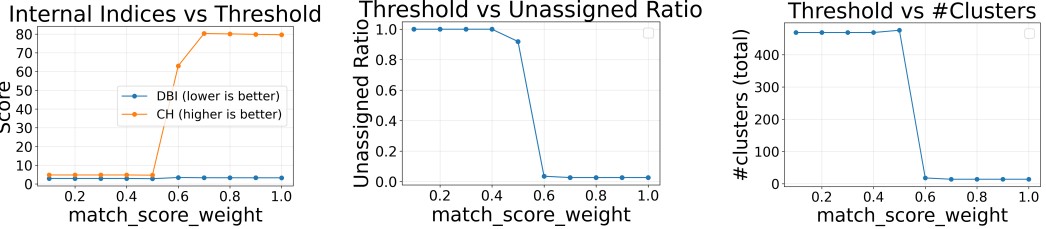

Figure 17: Match score weight sensitivity analysis for match cluster module.(left) Davies–Bouldin Index and Calinski–Harabasz Index for internal indices. (middle) Outlier Ratio for proper threshold. (right) Cluster Stability.

### C.3 DETAILS FOR LORA STORE MODULE

Table 1 shows that after filtering the atomized dataset, the fine-tuning effect has improved significantly. However, compared with the GSM8K dataset, improvements in the PUBMEDQA and LEXGLUE datasets are less pronounced. Below we describe how we process the three datasets so that subsequent researchers can reproduce our code and make further improvements.

For the PubMedQA dataset, we applied the following filtering conditions. The question must be causal/risk type and mention stroke or its subtypes. The evidence sentence must contain a causal trigger word and a normalizable object. The subject must be identifiable, with MeSH preferred; otherwise, heuristic extraction is used. The predicate must be extractable. The RESULTS paragraph is used first, followed by the long answer.

Table 2: Impact of Random Seed Variation on SAGE

| Seed | 123 | | | | 42 | | | | 7 | | | |
|------|-----|-----|-----|-----|-----|-----|-----|-----|-----|-----|-----|-----|
| | EM | MAE | MSE | NER | EM | MAE | MSE | NER | EM | MAE | MSE | NER |
| $Adp_1$ | 84.95 | 29185 | $1.5 \times 10^{10}$ | 78.58 | 98.28 | 11 | 39277 | 81.36 | 94.92 | 547 | $9.6 \times 10^7$ | 94.59 |
| $Adp_2$ | 99.80 | 0.05 | 1.36 | 99.40 | 95.63 | 1561 | $5.1 \times 10^8$ | 96.39 | 99.80 | 0.01 | 0.05 | 100 |
| $Adp_3$ | 99.80 | 0.01 | 0.16 | 100 | 99.60 | 0.05 | 1.36 | 100 | 98.99 | 4.56 | 5705 | 99.00 |

The filtering criteria for the LEXGLUE dataset include several steps. First, regular expressions are used to match questions involving causal or conditional judgments. Second, questions and options must contain core legal knowledge points of interest, such as "rule12b6" and "dismissal." Third, only sentences containing key trigger words or logical indicators—such as legal citations, case conclusions, or judges' arguments—are retained. Fourth, the subject or key object involved in the question is identified, such as "plaintiff," "defendant," or "court." Finally, actions or decision logic are extracted, such as "dismissed," "granted," or "violated."

The GSM8K dataset consists of math problems, whose format prevents causal reasoning similar to that performed on the PubMedQA and LEXGLUE text-based datasets. Therefore, we first extracted simple Level-1 math problems for training. The Level-1 extraction rules are as follows. "**" appears no more than 2 times, indicating fewer reasoning steps. "¡¡x=...¿¿" appears no more than once, indicating fewer inline calculations.

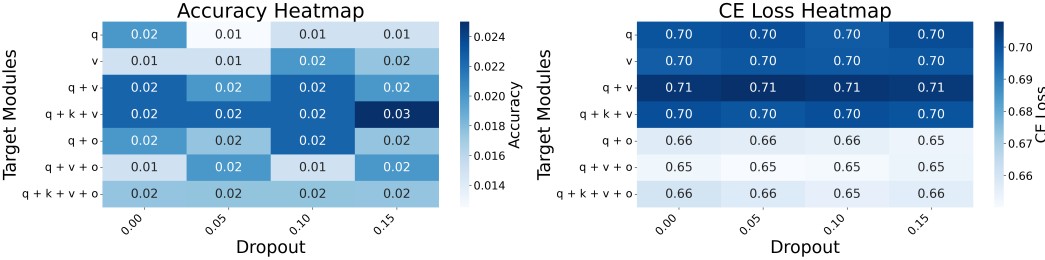

Figure 18: Heatmap of accuracy and cross-entropy loss for LoRA parameter: examining the necessity of exploitation optimization.

To further evaluate the LoRA Store module's performance in selecting nearest-neighbor hyperparameters for the refined adapter module (searching over rank and learning rate while keeping all other training parameters constant). we also experimented with nearest-neighbor search using other parameters and the results are shown in Figure 18. The accuracy and cross-entropy loss remain largely unchanged regardless of whether the target modules or dropout are modified. Specifically, the heatmaps for rank and learning rate reveal an accuracy difference range of up to 81%, whereas the corresponding range for target modules and dropout is only 20%.

### C.4 DETAILS FOR SAGE

To further assess the stability of SAGE, we tested the model by shuffling the data using different random seeds. The final results are presented in Table 3. The evaluation indicates that shuffling the data order has no significant impact on the performance of SAGE, demonstrating its high robustness to changes in data order.

## D. LIMITATION FOR SAGE

Our trigger module treats knowledge access as a filtering problem, where high accuracy and boundary recognition are critical. To this end, we incorporate ground-truth labels for auxiliary evaluation. In preliminary experiments on standard NLP benchmarks, this module achieved near-perfect accuracy ($\approx$100%) and did not require additional training compared to hidden-state approaches Chen & Varoquaux (2025). Although ground truth naturally exists for objective data (e.g., factual knowl-

edge), subjective domains such as sentiment lack explicit references. For such cases, surrogate strategies—such as weak supervision, self-consistency checks, or consensus-based labeling—may serve as alternative evaluation signals, broadening the applicability of the trigger design.

Beyond Trigger module, we propose atomized datasets for LoRA fine-tuning. We define atomization as the decomposition of complex reasoning chains into minimal independent steps, each corresponding to a single inference rule or transformation. For example, in mathematical reasoning, a multi-step proof can be atomized into elementary transformations such as algebraic substitution or logical implication. While our current implementation relies on human pre-selection of these steps, this process could be automated through curriculum learning or rule-based decomposition. This perspective resonates with curriculum learning and modular memory architectures Kang et al. (2025), yet it differs from retrieval-augmented generation Lewis et al. (2020), which relies on external retrieval; in contrast, our approach embeds atomized reasoning units directly into LoRA adapters. We view this as an initial step toward long-term memory formation, where integrating multiple atomized adapters could enhance LLM performance on complex reasoning tasks.

Currently, the SAGE architecture is primarily designed for NLP tasks. However, the scope of information that can be extracted from text alone is inherently limited, while a comprehensive memory system should encompass multiple modalities, including both textual and visual knowledge. Extending SAGE to multimodal inputs is therefore a natural next step. Fortunately, LoRA fine-tuning on image models has already become a well-established paradigm Yang et al. (2025); Farhadzadeh et al. (2025), and most image classification benchmarks provide clearly defined ground-truth labels. Nevertheless, aligning heterogeneous modalities introduces additional challenges, such as reconciling differences in feature space and balancing the contribution of each modality. This suggests that, during the initial phase of multimodal expansion, the alignment requirements for SAGE remain relatively modest. By leveraging these existing frameworks, SAGE can be readily adapted to handle richer information dimensions, paving the way toward a more general and scalable memory architecture. Our current evaluation is limited to NLP datasets, and we leave a systematic multimodal study for future work.

## E. THE USE OF LLMS

LLMs were used in a limited capacity to (i) suggest related work, which was manually verified by the authors for accuracy and relevance, and (ii) proofread the manuscript for grammar and tense. The models were not involved in research design, experiments, or interpretation. All scientific contributions and claims are solely the responsibility of the authors.

