# OpenReview forum: "SAGE: Adaptive Self-Training Towards Endogenous Memory in Large Language Models"
_ICLR.cc/2026/Conference — ICLR 2026 Conference Withdrawn Submission_

### Official Review · Reviewer_xB3c · 2025-10-15

**Soundness:** 2
**Presentation:** 2
**Contribution:** 2
**Rating:** 2
**Confidence:** 4

**Summary:**

This paper presents SAGE, a system to endow LLMs with a self-adaptive suite of components that allow incorporating new inputs with limited supervision. New inputs are filtered (to detect which ones are OOD, i.e., have not been used to pre-train the LLM) and clustered dynamically and in a streaming way (to separate new input into specific tasks). Based on a time-out, clustered data is used to finetune the LLM with efficient adapters, through an elaborate procedure to explore finetuning hyperparameters, and to select the best performing adapters for their integration within the LLM.

The main body of work is dedicated to SAGE evaluation. First, each system component is evaluated in isolation, to evaluate reliability in discerning in-ditribution vs. out-of-distribution data, to assess clustering quality, and to determine the effectiveness of "on-demand" fine-tuning using clustered OOD data vs. alternative baselines.
Finally SAGE is tested end-to-end, and compared to simple fine-tuning baselines.

**Strengths:**

* Timely topic: this work addresses the important topic of incremental LLM training under out-of-distribution assumptions. It is related to continual learning, and contributes to the literature in this domain.

* Compact and intuitive "systems" work: SAGE is clearly designed, each module contribution is assessed, and the functions executed by each module is sufficiently intuitive

* The experimental evaluation of SAGE components is appropriate, and the ablations are useful to determine the contributions of each module configuration.

**Weaknesses:**

* The editorial quality is a weak point. In my view, this is a systems work, and could have been explained in terms that involve minimal mathemtic notation. Indeed, this latter is one of the weakest aspect of this paper, because it is often inconsistent and hinders the task of understanding some important details. Since the function of each module of SAGE is intuitive, more space could have been given to better explain what is reported in Appendix D, under the limitations section.
Ultimately, the unnecessary display of equations (in my humble opinion) has a negative impact on a proper assessment of SAGE.

* In section 2, Sketch of SAGE, authors mention that to reduce adaptation complexity, they decompose complex reasoning tasks into atomic subtasks, which are minimal and independent. Nowhere in the main paper such a task is to be found! The reader has to delve into Appendix D to discover that this decomposition is a manual, human defined task. Even if one is ready to accept such practice as a preliminary step toward a fully autonomous system, this aspect should have been put forward and discussed and evaluated through ablations.

* The requirement for ground-truth labels is poorly discussed (except again, some details are offered in Appendix D). In lines 350-352, some information about the structure required by SAGE to operate on incoming samples is discussed, but I think this should have been clarified upfront.

**Questions:**

* In section 4.1: why do you use an external embedding model? Wouldn't the embedding layers of the base LLM constitute a natural, self-contained choice, as well as a challenge, since those layers would have been trained during pre-training, hence be sub-optimal for SAGE modules? Or is this the answer to my question :) ?

* The current SAGE design filters out in-distribution new samples. How is the LLM supposed to eventually "update" its internal knowledge in case of counter-facts? Imagine a mistake slipped in the pre-training data, stating that Picasso was a German painter. A new sample containing the correct country would not be considered OOD right?
More globally, SAGE seem to target exclusively OOD data integration rather than being a full system for LLM continual and incremental learning. I understand that this was clearly stated in line 36 of the paper, but what is the authors' take on this issue?

* I beg the authors to clarify what exactly is an input (figure 1). Is this a single token from a prompt, a full prompt, or the structured data type including: question, full prompt, real-answer, and label?

* Is SAGE designed for QA scenarios only?

* What are the reasoning chains that were used in the experiments? How come starting from the abstract throughout the introduction, SAGE is positioned to detect reasoning failures?

* What is the cost of SAGE? Every $T$ intervals, based on the trigger and trigger buffer outputs, it is required to finetune the base LLM with a large number of LORA variants (depending on hyperparameter choice). Is this realistic?

---

### Official Review · Reviewer_19n6 · 2025-10-26

**Soundness:** 2
**Presentation:** 2
**Contribution:** 2
**Rating:** 2
**Confidence:** 2

**Summary:**

This paper introduces SAGE, a framework for adaptive, test-time parameter updates in LLMs. The process is triggered when the model detects a reasoning failure on OOD knowledge. The framework consists of three modules: 1) a **Trigger** that identifies failures in real-time by comparing model outputs to ground-truth labels using confidence, behavior, and semantic metrics; 2) a **Trigger Buffer** that collects and clusters these failure samples using a streaming HDBSCAN-based algorithm to ensure stable, coherent data for training; and 3)  a **LoRA Store** that automatically managing a dynamic pool of optimized LoRA adapters. Experiments on a LLaMA-2-7B model, particularly on atomic subtasks from GSM8K, demonstrate high accuracy (97.16% EM) and robust knowledge updating.

**Strengths:**

1. **Clear Framework:** The paper addresses the critical problem of static LLM parameters. The proposed modular architecture (Trigger, Buffer, Store) is logical, clean, and presents a clear pipeline for test-time adaptation. The core concept of failure-triggered adaptation is highly intuitive.
2. **Data Curation Buffer:**  Instead of naively fine-tuning on every error (which could be noisy or catastrophic), the framework intelligently curates failure cases. By clustering samples and ensuring cluster stability using ARI and semantic coherence, SAGE likely improves sample efficiency and prevents model degradation.
3. **Dynamic Adapter Management:** The LoRA Store is a practical and effective solution for managing knowledge integration. The automated search for optimal LoRA hyperparameters and the maintenance of a dynamic adapter pool is a valuable contribution for efficient and continuous adaptation.

**Weaknesses:**

1. **Critical Dependence on Ground Truth**: The framework's most significant weakness is its reliance on ground-truth labels in the Trigger module (Section 3.1). This "oracle" is necessary to compute the Anomaly Score and detect failures. This assumption is unrealistic for genuine, real-world self-adaptation, which makes the "self-training" claim in the title questionable; the method is more akin to oracle-supervised, test-time fine-tuning on failure cases.
2. **Manual Task Decomposition:** The impressive results on GSM8K are heavily dependent on decomposing problems into "atomic subtasks." As noted in Appendix D, this decomposition relies on "human pre-selection." This manual, domain-specific feature engineering limits the framework's autonomy and generalizability. It is unclear how SAGE would function on complex, novel tasks that are not easily "atomized."
3. **Limited Scope of Evaluation:** The primary end-to-end evaluation is on GSM8K Level-1 tasks (1-2 computation steps). The framework's ability to scale to more complex, multi-step reasoning—where failure attribution is more difficult and atomic decomposition is more ambiguous—remains unproven.

**Questions:**

See weaknesses.

---

### Official Review · Reviewer_HUXL · 2025-10-30

**Soundness:** 1
**Presentation:** 1
**Contribution:** 1
**Rating:** 0
**Confidence:** 4

**Summary:**

The paper proposes SAGE, a three-stage test-time adaptation framework for LLMs that aims to build endogenous memory: (1) *Trigger* scores each prediction with four signals (logits margin, BLEU, ROUGE-L, embedding similarity) against the ground-truth answer and flags failures above a threshold.Then (2) *Trigger Buffer* collects failure cases and clusters them online. Finally (3), *LoRA Store* finetunes adapters per stable cluster. Experiments use LLaMA2 7B and four datasets: TriviaQA as ID, and PubMedQA, LexGLUE, GSM8K as likely OOD.

**Strengths:**

- The goal of endogenous memory in LLMs is important. If done right it would be impactful.
- Module-wise ablations and some sensitivity studies.

**Weaknesses:**

Overall, the paper is an over-engineered system that works under oracle conditions but does not establish a realistic label-free test-time adaptation method.

- simplistic assumptions on supervised training data availability (most critical): For example, the trigger uses BLEU, ROUGE-L, and embedding similarity to the gold answer. This assumes access to labels at test time. In real continual settings, labels arrive late, are sparse, or do not exist. The trigger then cannot run as designed. The current evaluation reduces to "if the model is wrong when we know the answer, fine-tune on that example," which is trivial.

- Unfair comparison and extra capacity:
SAGE adds many LoRA adapters and performs two-stage hyperparameter search. Baselines do not get an equal parameter or compute budget. A fair setup would either
a) constrain SAGE to the same total trainable parameters and search budget as the baseline (i don't see how) or
b) compare against mixture-of-adapters or adapter ensembles with matched total adapter parameters and a gating mechanism.
In practice, SAGE, at inference time, would use as many models (Base + Adapter) as clusters/adapters, which is much more capacity than a single model.

- Simplistic OOD detection: Beyond the oracle access to labels, the numeric signals are weak proxies. BLEU and ROUGE-L are poor fits for math answers. The logits margin normalization is ad-hoc. The claim of perfect ID vs OOD separation is not credible at the current stage.

- Clustering contribution is incremental: The buffer uses standard HDBSCAN with simple stability checks and merges. This is a reasonable engineering choice but thin as a core contribution. Many prior works already route among adapters or experts using embeddings.

**Questions:**

- Trigger without labels: How does the trigger work when gold answers are not available at test time? Which unsupervised or self-supervised signals can replace BLEU, ROUGE-L, and embedding similarity to the gold answer? how would performance be impacted?

- Gating vs MoE. How would your cluster-to-adapter routing compare to a mixture-of-adapters with a learned router trained under the same budget? What is the scaling behavior as the number of clusters grows.

Minors:
- it's the adjusted Rand index not Adjusted Random index
- line 409: citing wrong figure Fig18
- Please move metric definitions to an appendix. Use the space for more salient issues.

---

### Official Review · Reviewer_Lydi · 2025-10-31

**Soundness:** 1
**Presentation:** 3
**Contribution:** 2
**Rating:** 4
**Confidence:** 4

**Summary:**

This paper proposes to update the parameter weights of the LLM for adapting incrementally to new reasoning tasks. The methodology is divided in three steps. First, a trigger module detects the reasoning failures: given a question, this module scores a masked version of the ground truth answer, and considers it as out-of-distribution when the scores is below a certain value (meaning that the LLM answers differently w.r.t. the ground truth answer). Then, those failures are aggregated in streaming with clustering. Finally, the aggregated cluster of failures are used for tuning the parameters weights using a pool of LoRA adapters. This methodology is applied in real-time.

This methodology is applied on llama-2-7B using 4 datasets: triviaQA (this is considered in distribution), pubmedQA, lexGLUE, gsm8k (that are considered out of distribution). Results show a perfect separation of in- and out-distribution, and exact match of the fine tuned models of 97%.

**Strengths:**

- target the internal parameter weights, for which I agree will become a necessary step for modifying models' cognition
- idea that shares similarity with prototyping in online continual learning

**Weaknesses:**

- I'm not convinced that the trigger module is detecting a reasoning failure right now, the scoring seems closer to direct memorization. See also questions below.
- I have some doubt regarding the evaluation. The questions below could help to refine my current understanding
- the ground truth is needed for building the trigger module (this limitation is acknowledged in Appendix D)
- the scale of the experiment is small: the experiments should be performed on some more recent models, and probably on more realistic new tasks

**Questions:**

### Regarding the trigger module:

Q. How the methodology ensures that ID and OOD are correctly defined? Authors say that the pubmedQA (2019), lexGLUE (2021), gsm8k (2021) are "unlikely to be encountered during pretraining", while triviaQA (2017) "matches with the model's pretraining distribution". However llama-2-7B has been released in 2023, and the pre-training set is not publicly available.

Q. It is unclear that the trigger module captures different *reasoning* tasks. It might simply captures different answer format. For ensuring this, can you extend the in-distribution datasets?

Q. Is it needed to use all Eqs. 2,3,4,5 for separating ID vs OOD in your experiment? Given Fig. 11, the separation seems straightforward.

### Regarding the trigger buffer:

Q. Can you confirm that an input sample is a question/answer (according to l. 350) for the whole methodology? In particular, how is defined an anomalous sample that is clustered?

### Regarding the evaluation:

Q. Do you ensure that exact match performance on the initial knowledge is retained after fine tuning?

Q. The exact match performance is almost 100% after fine-tuning adaptation. Are you checking the performance on the training data or on a separate test data? The aim of the work, according to the abstract, is to adapt the reasoning part, not the memorization part.

### Minor questions/suggestions

CA-Lora is not defined (cluster augmented?)
- update some citations with the correct format
- l.154: Equation equation repeated
- Equation 1: $z_{(1)}$: this notation is referring to order statistics?
- Figure 7 and 8 should be labeled as Tables

Q. Can you give one sample example of Q/A for each of the dataset used?

Q. In Eq. 5, how w_i are selected? In Eq. 6, how tau is selected?

---

### Note · Authors · 2025-11-12

I have read and agree with the venue's withdrawal policy on behalf of myself and my co-authors.